# Immune Response to SARS-CoV-2 mRNA Vaccines in an Open-Label Multicenter Study in Participants with Relapsing Multiple Sclerosis Treated with Ofatumumab

**DOI:** 10.3390/vaccines10122167

**Published:** 2022-12-16

**Authors:** Tjalf Ziemssen, Marie Groth, Benjamin Ettle, Tobias Bopp

**Affiliations:** 1Department of Neurology, Center of Clinical Neuroscience, University Hospital Carl Gustav Carus at the TU Dresden, 01307 Dresden, Germany; 2Novartis Pharma GmbH, 90429 Nuremberg, Germany; 3Institute for Immunology, University Medical Center of the Johannes Gutenberg University, 55131 Mainz, Germany

**Keywords:** COVID-19 vaccination, relapsing multiple sclerosis, ofatumumab, neutralizing antibodies, T-cell response

## Abstract

Background: It is unclear whether multiple sclerosis (MS) patients receiving ofatumumab mount an immune response after SARS-CoV-2 mRNA vaccination. Methods: KYRIOS is an ongoing, multicenter, open-label, prospective clinical study on immune responses in MS patients after initial or booster SARS-CoV-2 mRNA vaccination prior to (cohort 1) or during (cohort 2) ofatumumab treatment. We report one-week and one-month results of the initial vaccination. A comparison with patients vaccinated while receiving beta-interferon, glatiramer acetate, dimethyl fumarate, teriflunomide or no treatment was included (cohort 3). Results: In total, 11 patients received their initial vaccination during the study. The primary endpoint of SARS-CoV-2-specific T-cells at month 1 was reached by 80.0% of patients in cohort 1 (N = 6) and 100.0% in cohort 2 (N = 5). T-cell reactivity peaked at week 1. All cohort 1 patients reached seroconversion for SARS-CoV-2 neutralizing antibodies at week 1 and month 1. In cohort 2, neutralizing antibodies increased in all patients and exceeded the cut-off for seropositivity in 40.0% of patients at week 1 and 25.0% at month 1. Immune responses in cohort 3 were comparable to cohort 1. Conclusion: Presence of T-cell response and increase in levels of neutralizing antibodies, although less pronounced compared to controls, suggest that MS patients receiving ofatumumab are able to mount an immune response to SARS-CoV-2 mRNA vaccination.

## 1. Introduction

Vaccination with the newly developed mRNA vaccines against the severe acute respiratory syndrome coronavirus 2 (SARS-CoV-2) efficiently protects healthy individuals against coronavirus disease 2019 (COVID-19) [1], but little is known about the efficacy of these vaccines in patients with multiple sclerosis (MS) treated with anti-CD20 therapies like ofatumumab.

Ofatumumab is the first fully human anti-CD20 monoclonal antibody authorized by the European Medicines Agency (EMA) for the treatment of adult patients with relapsing forms of multiple sclerosis (RMS) with active disease. After three initial weekly applications, it is applied subcutaneously every month and selectively depletes CD20^+^ B-cells, which represent one pillar of the adaptive immune response, but it does not affect CD20^−^ negative long-lived plasma cells that are responsible for protecting against reinfection with a pathogen [2].

The summary of product characteristics (SmPC) for ofatumumab suggests completing vaccine injections at least four weeks prior to treatment initiation for live or live-attenuated vaccines and, whenever possible, at least two weeks prior to initiation of ofatumumab for inactivated vaccines [3]. These recommendations are mainly based on the assumption that B-cell-mediated immunity is the major driver of vaccine-induced immunity. A humoral response has been considered to be the correlate of protection in conventional vaccination principles like influenza vaccines [4]. However, humoral immune response to conventional vaccines was attenuated in B-cell-depleted patients, as seen in the VELOCE trial on the anti-CD20 monoclonal antibody ocrelizumab [5], suggesting reduced protection through vaccination in these patients. Serum concentrations of immune globulin G (IgG) below the lower limit of normal are more frequent under ocrelizumab compared to ofatumumab, which suggests that subcutaneous application might be advantageous over intravenous application regarding the ability to mount an immune response to infectious agents and vaccinations [6,7].

The immune response to the newly developed SARS-CoV-2 mRNA vaccines has been shown to not only involve antibodies specific for SARS-CoV-2 but also functional CD4^+^ and CD8^+^ T-cells [8,9]. It has only been since recently that T-cell-mediated immunity after vaccination has been paid more attention. It has been shown that T-cells might be equally important, especially in vaccination against intracellular viral infections [10]. T-cells are a major mediator of innate and adaptive immunity and are of great importance for the immediate immune response. T-cells play an important role in the natural immune response of SARS-CoV-2 infection. It has been shown that specific CD4^+^ and specific CD8^+^ T-cells can be detected in the majority of COVID-19 patients [11], while some patients fail to establish seroconversion for specific antibodies [12]. Furthermore, it is known from other coronavirus infections that T-cell response was much more durable than B-cell response [13]. Immunological memory of SARS-CoV-2 has been recently demonstrated for eight months post infection [14]. Therefore, mRNA-based vaccination could potentially lead to an immune response and provide immunity through T-cell activation even in B-cell depleted patients like MS patients receiving ofatumumab.

The present study aimed to understand the impact of concomitant ofatumumab treatment on mounting cellular and humoral immune response after initial and booster SARS-CoV-2 mRNA vaccination, investigating the development of SARS-CoV-2-specific CD4^+^ and CD8^+^ T-cell responses as well as serum neutralizing antibody levels. Herein, we report the results of the immune response up to one month after the initial vaccination.

## 2. Materials and Methods

### 2.1. Study Design, Participants, and Treatments

KYRIOS is a prospective, open-label, two-cohort, multicenter study (EudraCT 2021-000307-20; NCT04869358) including RMS patients already treated or to be treated with ofatumumab as per physician’s discretion. Ofatumumab was administered as part of the study according to the SmPC. Patients had to be willing and eligible to receive SARS-CoV-2 mRNA vaccination, which was administered as part of clinical routine according to the SmPC.

Patients with a history of COVID-19 or current COVID-19 symptoms at screening, as well as patients who previously received a Bruton’s tyrosine kinase (BTK) inhibitor or an anti-CD20 therapy other than ofatumumab, were excluded.

Cohort 1 included patients receiving SARS-CoV-2 mRNA vaccination prior to starting ofatumumab treatment. Cohort 2 included participants receiving SARS-CoV-2 mRNA vaccination while already stable on ofatumumab treatment for at least four weeks since the first dose. Both cohorts were further divided into subcohorts according to their vaccination status. Accordingly, patients who had not yet been vaccinated against SARS-CoV-2 received their initial vaccination cycle during the study. The initial vaccination cycle comprises first and second vaccination, and optional additional booster vaccinations within the study were allowed. Patients who had already completed their initial vaccination cycle outside of the study only received their booster vaccination during the study. The response towards the initial vaccination in these patients was not part of the KYRIOS study.

An additional cohort (cohort 3) included patients vaccinated while on dimethyl fumarate (DMF), glatiramer acetate (GA), beta-interferons (IFN) or teriflunomide (TF) or while currently without disease-modifying therapy (DMT). Data on cellular and humoral response of patients in cohort 3 were available from the AMA-VACC study [15].

### 2.2. Outcomes and Assessments

The primary endpoint was the proportion of patients having established SARS-CoV-2-specific T-cells one month after the second dose of the initial vaccination cycle or after booster vaccination.

Secondary and exploratory endpoints included the proportion of patients with detectable SARS-CoV-2-specific T-cells at further timepoints, the proportion of patients with seroconversion with SARS-CoV-2-specific neutralizing antibodies, SARS-CoV-2-reactive T-cell levels and SARS-CoV-2 serum total and neutralizing antibody titers. Furthermore, the proportion of patients with confirmed COVID-19 after completion of the vaccination (at least 7 days after the second dose of the initial vaccination or after booster vaccination) was analyzed.

A comparison of SARS-CoV-2-reactive T-cell response and SARS-CoV-2 antibody response in cohorts 1 and 2 with the responses in cohort 3 was included as an exploratory endpoint.

SARS-CoV-2 reactive T-cells were measured by enzyme-linked immunosorbent spot (ELISpot) assay from peripheral blood mononuclear cells (PBMC) that were stimulated with SARS-CoV-2 peptide mix. The CoV-iSpot Interferon-γ + Interleukin-2 (ELSP 7010 strip format) from GenID was used, and each ELISpot assay was performed with 2 × 10^5^ PBMCs. Neutralizing antibodies were analyzed utilizing the cPass^TM^SARS-CoV-2 Neutralization Antibody Detection Kit from GenScriptUSA Inc., Piscataway, NJ, USA (L00847). SARS-CoV-2 serum total antibodies were detected by the Elecsys Anti-SARS-CoV-2 S immunoassay from Roche, which detects antibodies against the receptor-binding domain of the spike protein.

Assessments were performed at week 1, month 1, month 6, month 12 and month 18 after the second dose for patients who receive their initial vaccination during the study. For patients receiving only their booster vaccine within the study, assessments were performed at month 1, month 6 and month 12 after the booster vaccination. Additional vaccination is allowed at any time at discretion of the physician. One month after the first and second additional vaccination, an additional visit including blood sampling was be performed.

### 2.3. Administrative Procedures

The trial was conducted in accordance with the International Conference on Harmonisation guidelines for Good Clinical Practice and the principles of the Declaration of Helsinki. The protocol was approved by an ethics committee. All patients or their legal representatives provided written informed consent before commencing trial-related procedures.

### 2.4. Statistical Methods

The results of a pre-planned interim analysis of month 1 data are presented, i.e., after all participants have completed the study visit at month 1 after the second dose of the initial vaccination cycle or after the booster vaccination (data cut-off: 12 July 2022). This analysis constitutes the primary analysis of the study. Secondary data were examined as a preliminary evaluation of proof of concept along with relevant safety data.

No formal statistical testing was applied. All endpoints were analyzed descriptively. A target sample size of 20 participants per arm was selected based on the need for the early availability of results and the feasibility of recruiting sufficient participants.

For the primary analysis, the absolute numbers and the proportion of participants having established SARS-CoV-2-specific T-cells within each cohort are presented together with a 95% confidence interval (exact Clopper–Pearson). For secondary and exploratory endpoints, categorical data are presented as frequencies and percentages, and continuous data are presented as mean and standard deviation or median and range. Statistical analyses were performed with SAS version 9.2 (SAS Institute Inc., Cary, NC, USA).

## 3. Results

Among the included patients, 11 patients had their initial vaccination during the study. Of these, 6 patients received their initial vaccination prior to ofatumumab treatment (cohort 1), and 5 patients received their vaccination during stable ofatumumab treatment (cohort 2). Patient characteristics at the time of screening are shown in Table 1.

Briefly, mean age was 32.5 years in cohort 1 and 32.4 years in cohort 2, and disease history since diagnosis was 7.5 and 5.8 years, respectively. A total of 67% of patients in cohort 1 and 60% in cohort 2 were treatment-naive. Most patients received BioNTech/Pfizer SARS-CoV-2 mRNA vaccines, with an average of 3.0 weeks between first and second dose in cohort 1 and 3.4 weeks in cohort 2 (Table 2). In cohort 1, the first ofatumumab dose was applied one month (mean ± SD: 0.93 ± 0.02) after the second vaccination. The interval between the start of ofatumumab to the first vaccination was almost 2 months (mean ± SD: 1.85 ± 0.49). Characteristics of patients in cohort 3 (N = 20) for the exploratory comparison were published previously [15]. Briefly, mean age was higher with 51.0 years, and time since MS diagnosis was 9.1 years. In this cohort, 6 patients received glatiramer acetate, 3 patients received interferon-beta, 7 patients received teriflunomide, and 4 patients were not under DMT (Table 1).

The primary endpoint of SARS-CoV-2-specific T-cells at month 1 was reached by 80.0% patients in cohort 1 (initial vaccination prior to ofatumumab) and 100.0% in cohort 2 (initial vaccination during ofatumumab) (Figure 1A). At week 1, the responder rates were 100.0% in both cohorts, whereas in cohort 3 the responder rate was 60% (Figure 1A). ELISpot-based quantification of T-cell reactivity by calculating IFN-γ stimulation indices revealed a peak in T-cell reactivity at week 1 after full vaccination in all cohorts. At month 1, T-cell stimulation indices decreased compared to week 1 but remained higher than at baseline in cohorts 1 and 2 (Figure 1B).

All patients in cohort 1 reached seroconversion for SARS-CoV-2 neutralizing antibodies at week 1 and month 1 after initial vaccination. In cohort 2, seroconversion was reached by 40.0% at week 1 and by 25.0% at month 1, while in cohort 3, the seroconversion rates for neutralizing antibodies were 90% at week 1 and 95% at month 1. All patients showed an increase in neutralizing antibody titers as soon as one week after full initial vaccination, although titers remained lower in cohort 2 (Figure 2A). SARS-CoV-2 serum total antibody titers after initial vaccination are shown in Figure 2B. Seropositivity for SARS-CoV-2 serum total antibodies was reached by 100% in cohort 1, 60% in cohort 2 and 100% in cohort 3 at week 1. All patients with initial vaccination prior to ofatumumab initiation (cohort 1) and almost all patients in cohort 3 reached antibody titers of 250 U/mL after week 1 and month 1. Titers remained lower in patients receiving their initial vaccination during ofatumumab (cohort 2).

Until data cut-off (median observation time: 50.7 weeks), all 11 patients (100%) reported adverse events (AEs) during the study. Of these, five cases were DMT-related and five were vaccine-related (Table 3). One patient had experienced a relapse in cohort 1. In cohort 2, one relapse, which was classified as a serious adverse event, occurred before the 1st vaccination. The patient fully recovered. No further serious adverse events and no deaths occurred.

COVID-19 infections were reported in one patient in cohort 1 and three patients in cohort 2 until data cut-off. The infection in cohort 1 occurred at day 7 after the second vaccination. The infections in cohort 2 occurred at one month, two months and six months after the second vaccination. The latter patient had received a third vaccination two months before the infection. All infections were moderate according to the Common Terminology Criteria for Adverse Events Version 5.0 (CTCAE) grading (Grade 1: no intervention necessary; Grade 2: moderate symptoms; oral intervention indicated (e.g., antibiotic, antifungal or antiviral); Grade 3: intravenous antibiotic, antifungal or antiviral intervention indicated; invasive intervention indicated; Grade 4: life-threatening consequences; urgent intervention indicated; Grade 5: death), and all patients fully recovered. The duration of infection was 9 days in cohort 1 and from 10 to 13 days in cohort 2.

## 4. Discussion

According to the present KYRIOS data, SARS-CoV-2-specific T-cells could be detected in all patients vaccinated prior to ofatumumab treatment or during continuous ofatumumab treatment as soon as one week after the initial vaccination cycle and were still present after one month in all but one patient. Furthermore, the extent of T-cell reactivity was comparable between the ofatumumab cohorts and higher than in the older control cohort. This implies that ofatumumab treatment did not affect the development of SARS-CoV-2 specific T-cell response. All patients showed an increase in neutralizing antibodies. However, while all patients vaccinated prior to treatment with ofatumumab and almost all patients in the control cohort reached the assay-specific cut-off value for seropositivity at week 1 and month 1, the rates were considerably lower in patients vaccinated during continuous ofatumumab treatment. Taken together, all patients developed SARS-CoV-2-specific humoral or cellular response or both as soon as one week and still one month after the full initial vaccination cycle, irrespective of being vaccinated prior to or during ofatumumab treatment.

As far as T-cell response is concerned, a pooled analysis based on 12 datasets demonstrated no significant difference between MS patients on anti-CD20 treatment and untreated patients [16]. Analysis of T-cell reactivity in KYRIOS revealed a peak at week 1 and lower but remaining reactivity at month 1. It therefore seems reasonable to assess T-cell response early after vaccination [17]. However, in line with the present KYRIOS results, a recent study on the timing of SARS-CoV-2 vaccination in patients treated with ocrelizumab implies that anti-CD20 therapy does not reduce T-cell response. In the study on ocrelizumab, it has been shown that CD4^+^ and CD8^+^ T-cell responses were higher in MS patients vaccinated at early time points, i.e., one to three months after the last ocrelizumab application compared with MS patients not receiving DMTs [18].

Adaptive immune response to SARS-CoV-2 has been shown to involve IgG antibodies, as well as CD4^+^ and CD8^+^ T-cell reactivity [8,9,19,20,21]. Neutralizing antibodies, a subset of specific antibodies, have been shown to prevent the binding of virus particles to the host cells and to interrupt viral entry [22,23]. They are considered a more stringent correlate of protective immunity than total anti-SARS-CoV-2 antibodies [22]. The presence of neutralizing antibodies in combination with CD4^+^ and CD8^+^ T-cell reactivity have been proposed as suitable predictors of a protective immune response [24]. Therefore, the results of this study provide important information about whether ofatumumab-treated patients are able to mount potentially protective immunity after vaccination. It was obvious from KYRIOS data that antibody development is reduced when the vaccine is applied during ongoing ofatumumab treatment, as can be expected from a B-cell depleting anti-CD20-directed therapy. Accordingly, a meta-analysis including over 28 datasets with over 1000 MS patients on anti-CD20 therapy has revealed that humoral response to COVID-19 vaccine was significantly lower than in patients without DMTs (OR = 0.022, 95% CI: 0.015–0.031, *p* < 0.001). The seroconversion rate was reported to be reduced by 36.0%. The meta-analysis was mainly based on studies assessing humoral responses under rituximab or ocrelizumab after two doses of SARS-CoV-2 vaccines and included only 10 patients on ofatumumab. The authors found no difference in seroconversion rates in patients under rituximab compared to ocrelizumab, but a trend towards higher conversion rate in patients treated with ofatumumab compared to ocrelizumab. The analysis also revealed that vaccination within six months of the l week of anti-CD20 treatment application, mainly ocrelizumab and rituximab, resulted in lower seroconversion rates than vaccinations more than six months after the last application [16].

Mounting of immune response as assessed in this study is in line with clinical data from ALITHIOS regarding severity and duration of COVID-19 infections in ofatumumab-treated patients. Among 1703 patients receiving ofatumumab in the ALITHIOS study, 245 (14.4%) reported COVID-19-related adverse events (210 confirmed cases and 35 suspected cases). Among 476 fully vaccinated patients, the rate of breakthrough COVID-19 was 1.5% (7 patients). Of these cases, five were of mild to moderate severity, one was severe, and one was life-threatening. Among 74 partially vaccinated patients, 1.97% (11 patients) had COVID-19, including nine mild to moderate cases, and one severe case. All patients recovered [25]. This implies that vaccination effectively prevented infection. However, it has to be pointed out that infections reported from ALITHIOS occurred before September 2021, which implies that the immune escape variant omicron was not yet circulating [26]. In KYRIOS, no severe SARS-CoV-2 infections occurred in patients after initial vaccination, and the duration of the infection was 9 to 13 days. COVID-19 cases in KYRIOS were predominantly observed during early 2022, suggesting that at least partially they were due to the Omicron SARS-CoV-2 subtype [27]. Omicron escapes the immune response and leads to infections despite vaccination [28]. Nevertheless, the mRNA vaccines remain effective in preventing severe disease courses [29], which could also be observed in KYRIOS.

Four COVID-19 convalescent patients from ALITHIOS were analyzed for their humoral and cellular immune response. In line with the KYRIOS results, antibody response was impaired and could not be detected in three patients under continued ofatumumab. In one patient who had the last dose of ofatumumab 24 days prior to COVID-19 symptoms but had paused ofatumumab for 28 weeks beforehand due to low IgM levels, SARS-CoV-2-antibody titers higher than 250 U/mL had been detected. T-cell immunity was observed in three patients. In one patient, cellular response was not assessed due to vaccination before the scheduled T-cell assessment [30].

Despite the encouraging results, the KYRIOS study bears some limitations. Due to the small sample size, results have to be interpreted with caution and require further confirmation. The study was conducted before recommendations for an additional vaccination four weeks after the second dose were issued. Therefore, immune responses to the initial cycle were assessed after two doses of vaccine. Furthermore, at present final results are only available for one-month assessments. Data on the immune response after six months and after booster vaccination, respectively, will be presented separately as soon as the final data are available.

Overall, KYRIOS shows that patients receiving ofatumumab develop immune response after SARS-CoV-2 mRNA vaccination. While T-cell response is not affected by ofatumumab treatment, humoral response was reduced in these patients. This suggests that MS patients receiving ofatumumab are able to mount an immune response to SARS-CoV-2 mRNA vaccination. In order to ensure optimal response to SARS-CoV-2 vaccination, initial vaccination should preferably be completed before starting ofatumumab as proposed in the SmPC. However, according to the German Multiple Sclerosis Society (Deutsche Multiple Sklerose Gesellschaft, DMSG), delaying the initiation of DMTs due to pending SARS-CoV-2 vaccination should be weighed against the MS disease activity, and the individual’s risk assessment of infection with SARS-CoV-2 [31]. Initial vaccination should be complemented by an additional third application to reach higher antibody levels, as recommended by health authorities.

## Figures and Tables

**Figure 1 vaccines-10-02167-f001:**
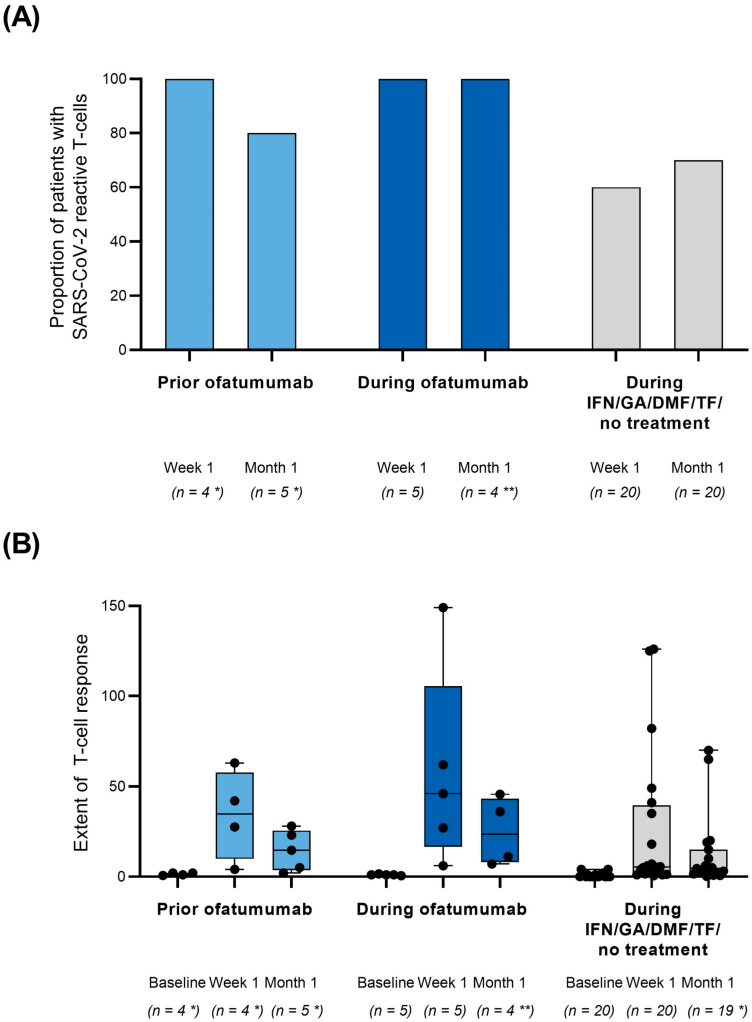
(**A**) SARS-CoV-2 specific T-cell responses after initial vaccination prior to or during ofatumumab treatment. (**B**) ELISpot-based quantification of T-cell reactivity after initial vaccination prior or during ofatumumab treatment by calculation of IFN-γ stimulation indices towards SARS-CoV-2 presented as mean (SD). * One patient in cohort 1 discontinued the study and only participated in a visit at month 1. For remaining patients, T-cell response could not be assessed due to technical problems. ** For one patient at month 1, visit could not be performed due to COVID-19 infection. DMF: dimethyl fumarate; GA: glatiramer acetate, IFN: interferon-beta; n: number of patients with assessments; TF: teriflunomide. Boxplots: min/max, horizontal line = mean. Black dots = individual values.

**Figure 2 vaccines-10-02167-f002:**
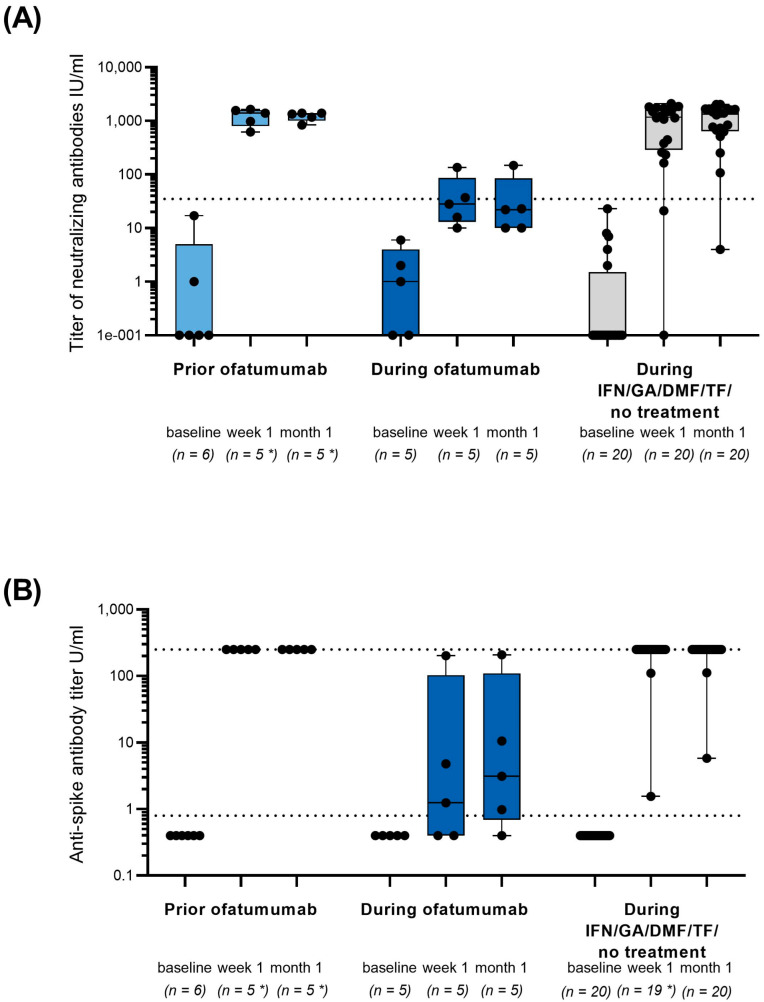
(**A**) Quantification of SARS-CoV-2-specific neutralizing antibody titer in U/mL after initial vaccination prior or during ofatumumab treatment. (**B**) SARS-CoV-2-specific serum total antibody titer in U/mL after initial vaccination prior or during ofatumumab treatment. * One patient in cohort 1 discontinued the study and antibody titers could only be assessed at baseline. Black dotted line indicates cut-off for seropositivity of the cPass^TM^SARS-CoV-2 Neutralization Antibody Detection Kit from GenScriptUSA Inc. (L00847) and Elecsys Anti-SARS-CoV-2 S immunoassay from Roche, respectively; gray dotted line indicates the maximal value of quantification range. DMF: dimethyl fumarate; GA: glatiramer acetate, IFN: interferon-beta; n: number of patients with assessments; TF: teriflunomide. Boxplots: min/max, horizontal line = geometric mean. Black dots = individual values.

**Table 1 vaccines-10-02167-t001:** Demographic and disease characteristics.

Variable *	Cohort 1 Vaccination Prior to Treatment	Cohort 2 Vaccination during Stable Treatment	Cohort 3 Vaccination during IFN/GA/DMF/TF/ No DMT
N	6	5	20
Age, years	32.5 (8.1)	32.4 (7.7)	48.6 (12.9)
Sex, female, *n* (%)	5 (83.3)	4 (80.0)	16 (80.0)
Time since diagnosis, years	2.7 (4.9)	1.5 (1.9)	13.99 (10.43)
Number of prior DMTs	1.0	0.4	1.6
Number of DMTs prior to ofatumumab, *n* (%)			
0	4 (66.7)	3 (60.0)	n.a.
≥1	2 (33.3) ^a^	2 (40.0) ^b^	n.a.

* If not indicated otherwise, data are presented as mean (SD). ^a^: cladribine and dimethyl fumarate ^b^: teriflunomide (2×). DMF: dimethyl fumarate; GA: glatiramer acetate, IFN: interferon-beta; TF: teriflunomide.

**Table 2 vaccines-10-02167-t002:** Vaccination characteristics.

Variable *	Cohort 1 Vaccination Prior to Treatment	Cohort 2 Vaccination during Stable Treatment	Cohort 3 Vaccination during IFN/GA/DMF/TF/No DMT
N	6	5	20
Vaccination, *n* (%)			
1st (BioNTech/Pfizer | Moderna)	5 (83.3) | 1 (16.7)	5 (100.0) | 0 (0)	19 (95.0) | 1 (5.0)
2nd (BioNTech/Pfizer | Moderna)	5 (83.3) | 1 (16.7)	5 (100.0) | 0 (0)	19 (95.0) | 1 (5.0)
Vaccination time interval, mean (SD)			
1st to 2nd vaccination, weeks/days	3.0 (0.1) weeks	3.4 (0.6) weeks	36.8 (9.0) days

* If not indicated otherwise, data are presented as mean (SD). DMF: dimethyl fumarate; GA: glatiramer acetate, IFN: interferon-beta; TF: teriflunomide.

**Table 3 vaccines-10-02167-t003:** Overview on adverse events.

Adverse Events, *n* (%)	Cohort 1 Vaccination Prior to Treatment (*n* = 6)	Cohort 2 Vaccination during Stable Treatment (*N* = 5)
Adverse events (AEs)	6 (100.0)	5 (100.0)
General disorders and administration site conditions	5 (83.3)	3 (60.0)
Nervous system disorders	5 (83.3)	4 (80.0)
Musculoskeletal and connective tissue disorders	1 (16.7)	2 (40.0)
Blood and lymphatic system disorders	1 (16.7)	0 (0.0)
Infections and infestations	3 (50.0)	4 (80.0)
Ear and labyrinth disorders	0 (0.0)	0 (0.0)
Gastrointestinal disorders	0 (0.0)	1 (20.0)
Injury, poisoning, and procedural complications	0 (0.0)	0 (0.0)
Metabolism and nutrition disorders	1 (16.7)	0 (0.0)
Psychiatric disorders	0 (0.0)	0 (0.0)
Reproductive system and breast disorders	0 (0.0)	0 (0.0)
Respiratory, thoracic and mediastinal disorders	2 (33.3)	1 (20.0)
Skin and subcutaneous tissue disorders	0 (0.0)	1 (20.0)
Vascular disorders	0 (0.0)	0 (0.0)
Not coded	0 (0.0)	0 (0.0)
AEs related to DMTs	2 (33.3)	3 (60.0)
AEs related to SARS-CoV-2 vaccine	3 (50.0)	2 (40.0)
AEs leading to permanent discontinuation of study medication	0 (0.0)	0 (0.0)
AEs leading to temporary interruption of study medication	0 (0.0)	1 (20.0)
Serious adverse events	0 (0.0)	1 (20.0)
MS relapse	0 (0.0)	1 (20.0)

In case of multiple AEs, a patient is counted only once in the respective category.

## Data Availability

Data will be provided upon reasonable request.

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
