# Peer review of "Immune Response to SARS-CoV-2 mRNA Vaccines in an Open-Label Multicenter Study in Participants with Relapsing Multiple Sclerosis Treated with Ofatumumab"

_vaccines, 2022, doi:10.3390/vaccines10122167_

Round 1
Reviewer 1 Report
Comment to the author
- It is an interesting study, reporting both cellular and humoral immune responses after COVID-19 vaccination in MS patients on ofatumumab treatment.
- Analysis of T-cell reactivity revealed a peak at week 1 and lower but remaining reactivity at month 1.
- For B-cell, all were vaccinated prior to treatment with Ofatumumab and almost all in the control cohort showed seropositivity at week 1 and month 1. The rates were considerably lower in patients vaccinated during continuous Ofatumumab.
I have some comments
- The immune responses depend on the baseline characteristics of the patients, time of testing, vaccine platforms, subtypes of COVID-19, methods of the assay used; T-cell i.e. CD4/CD8 activity, interferon-gamma release assay (IGRA), B-cell i.e. anti-spike antibody, a receptor-binding domain (anti-RBD IgG), neutralizing antibody (NAbs), inhibition percentage. Some studies also measure IgG levels for patients on anti-CD-20 therapy. Also, the efficacy of the vaccine could also measure as the proportion of patients having COVID-19 infection, grading, and severity of COVID-19 infection.
- Could you be more specific on
- patients on other DMDs before taking ofatumumab in Table 1, anyone was on other anti-CD-20 therapy or fingolimod?
- the detail in figure 2 for the cut-off point for neutralizing antibody titer and anti-spike antibody titer and reference the method of assay rather than describing “Black dotted line indicates assay-specific cut-off for seropositivity”
- What is an anti-spike antibody refer to? anti=RBD or just anti-spike-antibody.
- Does the author have data on the IgG level of patients after having Ofatumumab?
- Does the author have specific data on COVID-19 subtypes?
- Legend of Figure 1: Since the descriptions are the same for both 1A and 1B for “ *One patient in cohort 1 discontinued the study and only participated in visit at month 1 ** T-cell data for one patient is unavailable due to technical problems. ***For one patient at month 1, the visit could not be performed due to COVID-19 infection”.
The author may cut down the same sentences to ease the reader.
- The author report that 3 patients in cohort 2 got COVID infection and occurred at one month, two months, and six months after the second vaccination. Does the author have data on T-cell response, NAb, and anti-spike titer at the time of COVID infection?
- Page 10, line 308, Data on the immune response after six months "resp. " booster vaccination ............."resp. " should not be abbreviated.
Author Response
Dear reviewer,
On behalf of my fellow authors, I would like to thank you for the opportunity to revise and resubmit our manuscript vaccines-2085047 with its title “Immune response to SARS-CoV-2 mRNA vaccines in an open-label multicenter study in participants with relapsing multiple sclerosis treated with ofatumumab”.
We thank you for your thorough evaluation of our manuscript. We have carefully considered and responded to each suggestion. We have provided a separate point-by-point response. All changes to the manuscript are highlighted in the revised file using word track changes.
Point-by-point to reviewer's comments:
Reviewer 1
Comment 1
The immune responses depend on the baseline characteristics of the patients, time of testing, vaccine platforms, subtypes of COVID-19, methods of the assay used; T-cell i.e. CD4/CD8 activity, interferon-gamma release assay (IGRA), B-cell i.e. anti-spike antibody, a receptor-binding domain (anti-RBD IgG), neutralizing antibody (NAbs), inhibition percentage. Some studies also measure IgG levels for patients on anti-CD-20 therapy. Also, the efficacy of the vaccine could also measure as the proportion of patients having COVID-19 infection, grading, and severity of COVID-19 infection.
Could you be more specific on
Patients on other DMDs before taking ofatumumab in Table 1, anyone was on other anti-CD-20 therapy or fingolimod?
Response: We agree that the type of DMD before taking ofatumumab might have impacted immune responses. None of the patients had received anti-CD-20 therapy or fingolimod before ofatumumab. We have revised Table 1 to include details on prior therapy. In cohort 1, prior DMDs were cladribine and dimethyl fumarate, in cohort 2, bot DMD-pretreated patients had received teriflunomide (Table 1).
The detail in figure 2 for the cut-off point for neutralizing antibody titer and anti-spike antibody titer and reference the method of assay rather than describing “Black dotted line indicates assay-specific cut-off for seropositivity”
Response: We have included the assays used, which were cPassTMSARS-CoV-2 Neutralization Antibody Detection Kit from GenScriptUSA Inc(L00847) for neutralizing antibodies and Elecsys Anti-SARS-CoV-2 S immunoassay from Roche for anti-spike antibodies (lines 235 to 237).
What is an anti-spike antibody refer to? anti=RBD or just anti-spike-antibody.
Response: The assay used for quantification of anti-spike antibodies detects antibodies against the receptor-binding domain of the spike protein. We have revised the methods section accordingly (line 124).
Does the author have data on the IgG level of patients after having Ofatumumab?
Response: We agree that it would be interesting to have IgG levels of patients after receiving ofatumumab. Unfortunately, IgG levels were not assessed.
Does the author have specific data on COVID-19 subtypes?
Response: We agree that this would be interesting to know. Unfortunately, we have not assessed SARS-CoV-2 variants. We can only assume that COVID-19 cases in KYRIOS, which were pre-dominantly observed during early 2022, were at least partially due to omicron SARS-CoV-2 subtype. We addressed that assumption in the discussion section (lines 318 to 320)
Comment 2
Legend of Figure 1: Since the descriptions are the same for both 1A and 1B for “ *One patient in cohort 1 discontinued the study and only participated in visit at month 1 ** T-cell data for one patient is unavailable due to technical problems. ***For one patient at month 1, the visit could not be performed due to COVID-19 infection”. The author may cut down the same sentences to ease the reader.
Response: We have revised the figure and the figure legend accordingly, to make this easier for the reader (figure 1 and lines 189 to 196).
Comment 3
The author report that 3 patients in cohort 2 got COVID infection and occurred at one month, two months, and six months after the second vaccination. Does the author have data on T-cell response, NAb, and anti-spike titer at the time of COVID infection?
Response: Unfortunately, T-cell response, Nab and anti-spike-titers have not been assessed systematically at the time of COVID-infection.
Comment 4
Page 10, line 308, Data on the immune response after six months "resp. " booster vaccination ............."resp. " should not be abbreviated.
Response: We have revised the sentence accordingly (line 338).
Reviewer 2 Report
This study was done in a few patients followed for short time. The novelty is testing COVID vaccine response in MS patients after administration of ofatumumab (anti CD20 fully human monoclonal). Most of prior studies on response to SARS-CoV 2 vaccination were done in MS patients treated with a different anti-CD20 monoclonal called ocrelizumab. The authors conclude that they found presence of T-cell responses and of neutralizing antibodies suggesting that these patients are able to mount an immune response to SARS-CoV-2 vaccination (which is different from Ocrelizumab, were many patients treated with this anti-CD20 are not able to mount a response).
The manuscript is well written and the only addition I can ask is the that they should perform statistical analysis to see if there are any statistically significant differences for the parameters investigated.
As authors have mentioned the study has some limitations due to small sample size and short follow. In this context results have to be interpreted with caution.
Methodology, Conclusions and references are adequate.
Author Response
Dear reviewer,
On behalf of my fellow author, I would like to thank you for the opportunity to revise and resubmit our manuscript vaccines-2085047 with its title “Immune response to SARS-CoV-2 mRNA vaccines in an open-label multicenter study in participants with relapsing multiple sclerosis treated with ofatumumab”.
We thank you for your thorough evaluation of our manuscript. We have carefully considered and responded to each suggestion. We have provided a separate point-by-point response. All changes to the manuscript are highlighted in the revised file using word track changes.
Point-by-point to reviewer's comments:
Reviewer 2
Comment 1
The manuscript is well written and the only addition I can ask is the that they should perform statistical analysis to see if there are any statistically significant differences for the parameters investigated.
Response: We agree that statistical testing for significant differences in the investigated parameters would be helpful for interpretation. However, due to the small sample size, the study was not powered to test for significant differences, and only descriptive analysis can be presented. The results therefore have to be interpreted with caution.
Reviewer 3 Report
The authors describe the immune response to SARS-CoV-2 vaccination in a small group of patient. The immune response has been investigated through cell mediated and humoral immune response assays.
The introduction is exhaustive and well written, methods sounds and results are clear.
I have just two minor concerns:
- it is not clear the time interval between vaccination and ofatumumab in group one, and lat ofatumumab dose and vaccination in group two. please clarify
- COVID-19 severity has been described through CTCAE grading: in literature there are specific grading for COVID-19 severity, but CTCAE is correct in this context. I just suggest to add the explain the abbreviation and add a bibliography.
The paper is very novel and reader's interest in this field is high: I suggest to publish the paper with priority.
Author Response
Dear reviewer,
On behalf of my fellow authors, I would like to thank you for the opportunity to revise and resubmit our manuscript vaccines-2085047 with its title “Immune response to SARS-CoV-2 mRNA vaccines in an open-label multicenter study in participants with relapsing multiple sclerosis treated with ofatumumab”.
We thank you for your thorough evaluation of our manuscript. We have carefully considered and responded to each suggestion. We have provided a separate point-by-point response. All changes to the manuscript are highlighted in the revised file using word track changes.
Point-by-point to reviewer's comments:
Reviewer 3
Comment 1
It is not clear the time interval between vaccination and ofatumumab in group one, and lat ofatumumab dose and vaccination in group two. please clarify
Response: In cohort 1, the first ofatumumab dose was applied one month (mean±SD: 0.93±0.02) after the second vaccination. The interval between the latest dose of ofatumumab and the first vaccination was not documented. Instead, we calculated the interval between the start of ofatumumab treatment to the first vaccination, which was almost 2 months (mean±SD: 1.85±0.49). We have revised the results section accordingly (lines 164 to 166).
Comment 2
COVID-19 severity has been described through CTCAE grading: in literature there are specific grading for COVID-19 severity, but CTCAE is correct in this context. I just suggest to add the explain the abbreviation and add a bibliography.
Response: We have revised the manuscript and added an explanation of the abbreviation as well as a footnote explaining the classification. Accordingly, CTCAE classification is as follows. Grade 1: no intervention necessary; Grade 2: moderate symptoms; oral intervention indicated (e.g., antibiotic, antifungal, or antiviral); Grade 3: intravenous antibiotic, antifungal, or antiviral intervention indicated; invasive intervention indicated; Grade 4: life-threatening consequences; urgent intervention indicated; Grade 5: death (lines 255 and 256; footnote 1).